# Understanding transnational healthcare use in immigrant communities from a cultural systems perspective: a qualitative study of Dutch residents with a Turkish background

Aydin Şekercan,[1] Janneke Harting [ID],[1] Ron J G Peters,[2] Karien Stronks[1]

**To cite:** Şekercan A, Harting J, Peters RJG, *et al.* Understanding transnational healthcare use in immigrant communities from a cultural systems perspective: a qualitative study of Dutch residents with a Turkish background. *BMJ Open* 2021;11:e051903. doi:10.1136/bmjopen-2021-051903

¹Public and Occupational Health, Amsterdam UMC, Location AMC, Amsterdam Public Health research institute, Amsterdam, The Netherlands
²Cardiology, Amsterdam UMC, Location AMC, Amsterdam Public Health research institute, Amsterdam, The Netherlands

**Correspondence to**
Dr Janneke Harting;
j.harting@amsterdamumc.nl

## ABSTRACT

**Objectives** Transnational utilisation of healthcare by people with an immigrant background carries risks, including medicalisation and adverse iatrogenic outcomes. We investigated the drivers behind such transnational healthcare use from a cultural perspective on health systems.

**Design** Qualitative interview study (2018).

**Setting** Two primary care practices in Amsterdam, the Netherlands.

**Participants** Thirteen Dutch patients of Turkish background, who had obtained healthcare in Turkey, and who in general visited the primary care practice more than once a month.

**Results** In the respondents' stories, we observed how: (1) cross-border healthcare use was encouraged by cultural mismatches between expected and provided services and by differing explanatory models of illness upheld by patients and Dutch providers; (2) both transnationalism in patients and entitlements to insurance reimbursement facilitated the use of Turkish health services to bypass perceived barriers in the Dutch system; (3) cultural mismatches were reinforced during general practitioner consultations after the patients' return to the Netherlands, thereby inducing further service use abroad.

**Conclusions** Although cultural system influences are difficult to bridge, measures to reduce the unwelcome consequences of transnational healthcare use may include (1) strengthening the provision of culturally sensitive care in the country of residence and (2) restricting the reimbursement of care in the country of origin while maintaining the option to obtain care abroad.

## BACKGROUND

Across the globe, the transnational ties of former migrants and their families are enabling them to retain and pursue private, cultural and economic interests in their countries of origin, including the domain of healthcare. This phenomenon is known as transnationalism.[1 2] People with an immigrant background tend to use similar healthcare services in both their country of origin

---

and their country of residence.[1 3] High transnational healthcare use may increase the risk of medicalisation, defined as expansion of the medical domain.[4] Different diagnostic approaches may cause more ailments to be labelled as diseases[4] and thus also increase the risk of iatrogenic complications resulting from additional treatments.[5 6] Due to a lack of cross-border medical information transfers, on the patients' return to their country of residence they may be subject to repetitions of diagnostics, drug interactions in treatments and inadequate treatment aftercare.[7] In addition, transnational healthcare use is likely to foster antimicrobial drug resistance.[6] At the system level, it may increase healthcare spending in the country of residence, pull resources from the public sector in the country of origin, and thus widen health disparities.[6] Motivators for treatment abroad generally include low costs, short waiting lists, quality of care and the available medical

**BMJ**

procedures.[8 9] Yet, the drivers that underlie frequent transnational healthcare use are complex and deserve further investigation,[9] as to curtail its adverse effects.

People of Turkish ethnic origin who are resident in the Netherlands are a relevant group for further study of this phenomenon. With about 425 000 people, they form the largest ethnic minority group in the Netherlands.[10] Migration from Turkey was encouraged in the 1960s and early 1970s to fill labour shortages in unskilled occupation.[11] In the years that followed (1970–1980), many 'guest workers' brought their spouses and children to the Netherlands.[11] As 80% of the Dutch residents of Turkish background identify as being both Dutch and Turkish and maintain ties with the Turkish culture, they can readily compare, use and evaluate various features of health services in both countries.[2 7] Being residents of the European Union, they have access to necessary healthcare in all countries in Europe[12] and Dutch insurance companies reimburse public as well as private healthcare in both the Netherlands and Turkey.[13] There is also evidence that succeeding generations may continue to use healthcare in Turkey in the future.[12 13] Studying Dutch residents of Turkish background also has international relevance in view of the large ethnic Turkish populations in other northern European countries, including Germany, France, Austria, Belgium and Denmark.[10 14]

Transnational health service use is highly prevalent in Dutch residents with a Turkish background. They are more likely to use health services in the country of origin than other migration groups—46%, as compared with 18% of residents with a Moroccan background.[15] This happens even though apparently equivalent facilities are available in the Netherlands.[16] Moreover, those who frequently use health services in Turkey also tend to be frequent healthcare users in the Netherlands (ie, often in the top 10% of frequent users). Their overall rates of utilisation are also high in comparison with their coethnics who do not obtain healthcare in Turkey (Dutch primary care use: 40.4% vs 22.5%; specialist medical care use: 57.8% vs 33.9%).[15 17] The few factors identified to be associated with healthcare use in the country of origin are poorer self-reported health and a wider perceived cultural distance to the healthcare system in the country of residence.[16]

Previous research has primarily examined the immediate motives reported by people for using cross-border healthcare.[1] From an individual perspective, former migrants appeared to use services in the country of origin on opportunistic occasions, persuaded and guided by their social networks or spurred by a high burden of health problems or unmet needs attributable to language or cultural barriers.[3 16 18] However, some are also driven by factors relating to both national healthcare systems. They report, for instance, long waiting times, unnecessary delays, limited access to specialist care in their country of residence,[3 16 18] and rapidity and effectiveness of services in the country of origin.[16 19] Together, experiences like these act as push and pull factors for cross-border healthcare

use.[19] Studies among people using services in different healthcare systems addressed either the use by migrants of traditional or complementary medicine to complement conventional services within a Western country, or the use of traditional medicine to complement conventional medical services within a non-Western country.[20 21] We are not aware of studies that have used the cultural systems perspective to explore in depth the experiences that people with an immigrant background have with utilising conventional healthcare services in both the country of residence and the country of origin.

To better understand the motivations and mechanisms that underlie frequent service use in the Dutch and Turkish healthcare systems by Dutch residents of Turkish origin, we designed a qualitative study guided by the cultural systems perspective as proposed by Kleinman.[22] Kleinman regards each healthcare system as a cultural system with its own set of rules and values for dealing with health, illness and healing, even though different systems may share the same foundation of conventional medicine. Such a cultural systems perspective could help us understand the compatibility between a healthcare system and the cultural values of its users by comparing the different systems during the actual transnational use of healthcare by patients.

## METHODS

### Analytical framework

Viewing a healthcare system as a cultural system enabled us to compare its cultural rules with the cultural values of its users.[22 23] Differences therein express themselves during healthcare consultations as mismatches between the patient's and the provider's explanatory models of illness. Such models include beliefs about aetiology, symptoms, pathophysiology and course of sickness and/or treatment. In the case of transnational healthcare use, cultural mismatches between explanatory models are more likely to occur, as a medical consultation is influenced by the rules and values of two national healthcare systems and two national cultures. Therefore, the analytical framework comprised the concept of cultural mismatches, the differences between national healthcare systems and a comparison of national cultures.

Cultural mismatches may occur in all three phases of a consultation.[22 24] In the first phase, a patient presents his or her sensations, as the basis of the recognition that something is wrong, and tries to persuade the healthcare provider to transform these sensations into symptoms, as a recognised objective clinical reality.[24] Core to the second phase is the transition of an illness into a disease, once the healthcare provider, through further diagnostics, links the patient's symptoms to a particular disease, syndrome or condition. In the final phase of the consultation, the provider presents a treatment regimen. Each consultation phase thus leads to a specific endpoint: defined symptoms, diagnosis and treatment. These endpoints

may be—or may not be—acceptable within the patient's explanatory model of illness.

If we compare the Turkish and Dutch healthcare systems, they appear to generally provide similar quality of care in objective terms, but they differ in the organisation and delivery of services.[25] In Turkey, alongside municipal health centres for low-complexity non-acute care, patients have direct access to specialist care.[26] In addition, the Turkish healthcare system is interventionist in nature, including a generous use of both the available diagnostics (aimed at 'ruling in' disease) and the available treatment options.[27] In the Dutch system, general practitioners (GPs) provide low-complexity care, while also acting as gatekeepers in deciding if and when referral to specialist care is needed.[26] Dutch primary care is non-interventionist in nature, reflected by both its stepwise diagnostics (aimed at 'ruling out' disease) and its stepwise treatment. This wait-and-see approach results in a parsimonious use of screening, diagnostics and drug prescription.[26]

In terms of national cultures, Turkey has a more hierarchical society than the Netherlands.[28] In such a society, power, authority and control are often centralised (physicians make the decisions, eg). In addition, collectivity is more important in the Turkish than in the Dutch culture.[28] That may imply, for instance, that the family is more involved in the healthcare utilisation trajectory and that the provision of care is more family based. A final difference in national cultures is the higher level of uncertainty avoidance in Turkey.[28] This means that members of the society feel a greater degree of discomfort with uncertainty and ambiguity. This may translate into a stronger desire to rule out health risks and potential diseases and to rule in some pathophysiological cause to explain sensations or symptoms. Despite these differences between the two nations, it should be noted that culture is a dynamic concept, that may express itself in different ways.

These differences between the Turkish and Dutch healthcare systems and national cultures create a high potential for cultural mismatches in the medical consultations of transnational patients, as they influence which endpoints are reached and which are considered acceptable in medical consultations. If left unaddressed, cultural mismatches may heighten the risks of medicalisation and of iatrogenic reactions.[23 24]

### Participant recruitment, selection and setting

Two primary care practices in Amsterdam joined the study. Both practices included an ethnic-concordant GP, thereby excluding poor mastery of Dutch as a potential reason for healthcare use in Turkey.[1] Patients were identified by their GP as candidate respondents if they came in for consultation after obtaining healthcare in Turkey, and if they generally visited the Dutch practice more than once a month. The GPs were asked to invite a variety of candidates to ensure diversity in terms of gender, education and migration generation. After being notified by the GP, AŞ phoned sixteen Dutch primary care patients

with Turkish a background for an interview appointment. Three patients declined participation due to work obligations at possible interview times.

### Data collection

The interviews took place from August to November 2018 in a private room at the primary care practice or at the patient's home, depending on their preference. Some respondents wanted their spouses and/or children to be present in order to help to recollect, supplement and/or clarify their own stories. AŞ made sure that the presence of others was voluntary. The interviews ranged from 45 min to 120 min. They were held primarily in Turkish, although some eventually became mixtures of Turkish and Dutch. All interviews were audiotaped and, during transcription, translated into English by AŞ.

After the interview, a semistructured questionnaire was administered to assess the healthcare consumption in Turkey and sociodemographic aspects, including self-reported difficulty with the Dutch language.

### Interview approach

The biographic-narrative interpretive interview method was used, as it is well suited for retrieving enriched data in the domain of healthcare.[29] AŞ started the interview with the following request: 'Please tell me the story of your life, in the sense of all the events and experiences that have been important to you personally regarding health, sickness and seeking care.' In the first three interviews the respondents tended to focus on one episode of healthcare use or to discuss different episodes while mixing these up. Therefore, in the following interviews the interviewer began showing respondents a diagram of a human life cycle, so as to visually guide them in retrieving their stories in a chronological order without interfering in their thought-forming processes. In the second part of the interview, while sticking to the respondent's order of topics raised and their choice of words used, the interviewer asked the respondent to elaborate on or clarify certain narratives, specifically in relation to the different phases of the consultation. For the interview guide, please see online supplemental material 1.

### Data analysis

In the first phase of the data analysis, AŞ and JH familiarised themselves with the content of the data by reading through the interview transcripts. Next, they composed an initial coding scheme, inspired first by the interview method and grounded theory principles, and second by the above analytical framework and the literature on cross-border care. In the subsequent phase, AŞ coded the first four interviews. JH read the coded interviews and suggested changes to the coding scheme and/or coded segments. These suggestions were discussed and changes were made by consensus, resulting in a final coding scheme (see online supplemental material 2). Using qualitative data analysis software (MaxQDA, V.2018, VERBI, Berlin), the first author coded all interviews, which were

then checked by the second author. Emerging themes and patterns were visualised and discussed together. The pattern of drivers behind transnational healthcare use was almost homogeneous across the sample. Both authors deemed data saturation to have been reached after analysing nine interviews.

### Informed consent
Respondents were asked to sign a written consent form for the use of their recordings (for this study only), to confirm their ownership of their recordings at all times and to guarantee confidentiality.

### Patient and public involvement
No patients or public involved.

### Reflexivity
AŞ had extensive training in qualitative research and previous experience with the same community and topic.[7] During the interview introduction, he explained his role as both a researcher and a physician. It appeared that respondents tended to consider him a member of the same community who had been successful in developing his potentials in Dutch society. Respondents also seemed proud that he was benefiting the community due to his choice of a medical career. AŞ was aware that such ascribed authority could lead to socially desirable reporting. He attempted to avoid that by creating an open atmosphere and by staying alert to hesitations in respondents' narratives and any disagreements with his summarisations of their stories. Although all respondents appeared willing to participate in an open and honest fashion, a few initially showed signs of distrust due to the ethnic concordance and the medical profession of the interviewer. They seemed wary of their stories being shared with community members or other health professionals. After reassurance about the anonymity of

respondents and their option of halting the interview at all times, all respondents were eventually willing to openly share their intimate narratives.

## RESULTS
After describing the respondents' characteristics (table 1), we highlight the prominent themes that emerged in their narratives on (1) their medical consultations in the Netherlands, (2) their healthcare utilisation in Turkey and (3) their subsequent visits to their Dutch GP.

### Respondent characteristics
Our 13 participants were between 39 and 78 years of age; 5 were in paid employment, 6 were born in the Netherlands and 5 reported difficulty with the Dutch language (table 1). Their primary reason to go to Turkey was visiting relatives, while decisions about using healthcare were in general made after arriving in Turkey. Respondents usually made use of private healthcare services, as these were regarded more accessible and providing quicker services than public services.

### Healthcare utilisation in Netherlands
#### Illness presentation and persuasion
In the first phase of consultation, respondents typically felt that their GP chose a wait-and-see approach, and they often did not understand why. They felt that their sensations were either not acknowledged as symptoms, or were interpreted as symptoms giving insufficient reason for further diagnostic investigation to rule out underlying causes. This approach was out of line with the perceived urgency of the respondents' embodied sensations and their suspicions that something was wrong. All but one of the respondents explained that both basic physical examination and lab testing were definitely needed to

| Table 1 | Characteristics of interviewed respondents | | |
|---|---|---|---|
| | **Summarised characteristics** | | |
| Participants (no) | 13 | | |
| Age (range) | 39–78 | | |
| Gender (% female) | 62 | | |
| Migration generation* (% first, 1.5, second) | 31 | 23 | 46 |
| Employment status (% paid, % unemployed, % retired) | 38 | 54 | 8 |
| Educational attainment level† (% low, middle, high) | 70 | 30 | 0 |
| No of years in NL (% born in NL, range) | 31 | 24–54 | |
| No of yearly visits to TR (range) | 0.5–4 | | |
| Preference to stay in which country‡ (% NL, % TR, % circular) | 23 | 23 | 54 |
| No of diseases (range) | 0–6 | | |
| Difficulty with Dutch language (% yes) | 31 | | |

*1.5 migration generation: migration to NL before age 12.
†Low: primary or less education; middle: lower general or vocational secondary; high: upper general or vocational secondary or tertiary.
‡Circular: preference to stay longer periods of time in both countries.
NL, Netherland; TR, Turkey.

Şekercan A, et al. BMJ Open 2021;**11**:e051903. doi:10.1136/bmjopen-2021-051903

find out the cause (Q01). The vast majority said they tried numerous times to convince their GPs to deviate from the wait-and-see approach, often without success (Q02).

Q01: My first experience…. I was at work, I'd been tired lately, I only wanted to sleep. I went to my GP. He was my first GP after my own GP retired. He was new, fresh. He didn't even do a physical examination. His answer was, 'Yeah, just keep going. You're young, you don't want to be on social welfare.' I said, 'You don't know me. You didn't even do a medical examination and you don't know what I have.' He told me, 'You should just live with it, there's some things we can't explain.' I said to him, 'What did you do? Nothing.' I went away angry.

[I-03, 1rst generation migrant, paid work, no difficulty with Dutch language]

Q02: This time with my right leg…. Four months ago I had a sudden pain in my right leg. I went to my GP; the pain was of a degree that I couldn't walk. I was stumbling. I went to my GP and said my leg really hurt, it wasn't a normal pain. I couldn't stand, I had to lean on my knee to vacuum the floor. I said to him that the pain in my leg was not a normal pain. He said no, you probably overstrained it. I went to my GP twice, and the third time I went to the emergency department. I went to the … hospital; on the phone they'd said not to worry about it, just take [painkiller] and it'll go away. I went to the GP for the third time with my daughter *[anger in the voice]*. I said to him there's a torn tissue in my foot, I'm not a doctor but a patient. He didn't believe I had the pain. My … GP said that if it was torn there would be a bruise. I said this is not a normal pain. He gave me [painkiller] again.

[I-01, 1rst generation migrant, paid work, no difficulty with Dutch language]

Respondents frequently indicated that the wait-and-see approach did not provide them with an acceptable endpoint. It made them feel they were not taken seriously or not getting recognition as a patient. Most wondered whether they were treated differently to ethnic Dutch patients, and some explicitly insisted that they were. Some attributed the wait-and-see approach to language barriers, but others interpreted it as being treated as a second-class citizen (Q03).

Q03: If it really is a doctor who is discriminating, then you see the attitude immediately. He calls your name and walks back. *[By contrast:]* 'Mister Janssen' *[typical Dutch surname]*…. He waits. 'Hello Mister Janssen, I'm Dr Smit, please come with me.' Stays politely behind that man. The doctor stays behind that man. But when it's a foreigner, whoosh…. You don't see where the doctor's gone to.

[I-05, 2nd generation migrant, paid work, no difficulty with Dutch language]

### Illness-to-disease transition

In the second phase of the consultation, even if sensations of illness got transformed into symptoms, respondents typically insisted they still needed proof that the symptoms had a pathophysiological origin. They often did not understand why their Dutch GP failed to use all the available diagnostics to further 'rule in' a cause (Q04).

Q04: What kind of testing did you guys do? … If I have a problem, the answer is take a paracetamol, see how it's going a week later and then we'll check further. A week later, if it didn't work, they do some lab tests, but it's only one or two options they look at. No findings, then again, and again and again. It's tiresome…. Nowadays we have everything, the best machines, but if they're not used properly then what good are they to me?

[I-03, 1rst generation migrant, paid work, no difficulty with Dutch language]

Generally, stepwise diagnostics did not address the respondents' need for risk avoidance and their feeling that something bad could be happening. Especially cancer was believed by almost all patients to be unpredictable in its presentation, pace and mortality, and should therefore be ruled in with all the tests available. Hearsay experiences in their social network, where people believed unnecessary harm or death could have been prevented, reinforced the pressing need most patients felt to follow-up on their symptoms (Q05).

Q05: Why does it take a month in the Netherlands [to get an appointment]? You get cancer. A friend tells you he didn't hear about his for 2 months. Why can't [the testing] be the next day? But it takes a whole month. It might've metastasised by then. These events happened recently, what more can I tell you? There's lots of stories like that. Because I often go as a translator I know what those people experience.

[I-09, 2nd generation migrant, unemployed, no difficulty with Dutch language]

In such cases, most respondents did not regard the diagnosis offered to be an acceptable endpoint but believed referral to a specialist was needed. However, given the gatekeeper role of Dutch GPs, respondents reported they did not get referred easily. While a few respondents believed that certain authorities were prohibiting specialist referral, the GP's reluctance made most respondents question the medical knowledge of their doctor (Q06). As a result, some respondents said they felt part of an experiment, having to undergo a variety of tests first before receiving proper diagnoses (Q07).

Q06: I don't blame the GPs, because they only have certain knowledge. But then I say okay, I can accept that you only possess certain knowledge, I can appreciate that, but if it's beyond your knowledge then send me to a specialist…. If you don't have the knowledge, then you need to refer me.

[I-05, 2nd generation migrant, paid work, no difficulty with Dutch language]

Q07: She [the GP] told me it's probably gastroenteritis, even though I'd told her I hadn't been eating or drinking for a couple of days and felt like knives were being stabbed in my gut. She didn't do anything. She told me I had to submit a stool sample the following day to the lab, so they can maybe see it's a bacteria. The following day I didn't end up at the lab but in the emergency department, because I fainted. They brought me to the emergency department and they found I was completely dehydrated. The GP hadn't done any checks. I was in hospital for 2 weeks.

[I-03, 1rst generation migrant, paid work, no difficulty with Dutch language]

### Acceptable treatment regimen

All respondents' stories reflected a conviction that when something is wrong it needs a quick fix. Hence, they did not understand why they needed to try different treatments first before they got the most optimal treatment available that would eradicate their symptoms (Q08). In the respondents' perceptions, being given mere symptom relief instead of treatment for the underlying cause (which they assumed to be present) was no acceptable treatment procedure and just caused unnecessary delay (Q09). Again, some respondents said they felt like part of an experiment in which other treatment options were tested out before effective treatment was provided (Q10).

Q08: For example, I tell them I have stomach aches or that something else hurts. Maybe there's a bacteria. However, they want to do some research [with different treatments] first. First they'll give you a painkiller to try, and if that doesn't work than we can try another one. If you've already had a light medication, why don't they just try prescribing the other medicine that will have an effect?

[I-08, 2nd generation migrant, unemployed, difficulty with Dutch language]

Q09: They only thing they can tell you and do for you is give painkillers and paracetamol, nothing else. In the Netherlands, how can I say it, they only take care of your disease when it has progressed for quite a while.

[I-12, 1.5 generation migrant, unemployed, no difficulty with Dutch language]

Q10: I got a splint that would slip off after two steps. I went back and got another one. Exactly the same, after two steps it slipped off. I went back again. Listen, what are we up to here? What do you guys want? Am I the first person this happened to, that had to walk around with a splint? After great difficulty … I got referred…. Listen, why didn't you guys give me that in the first place? Why? It was a mere €10 more…. They don't come through with the full treatment

but do everything step by step. I don't mind that. I can understand that they don't give my body an uppercut. I can understand that, but some things like those splints – why do you need to treat me like an experiment?

[I-05, 2nd generation migrant, paid work, no difficulty with Dutch language]

### Healthcare utilisation in Turkey

Provoked by their experiences in the Dutch healthcare system (including perceived lack of recognition, mismatches in explanatory models of illness and the provision of unaccepted diagnoses), the majority of the respondents felt a strong need to consult the Turkish system, which they typically effectuated during an already scheduled holiday.

In Turkey, respondents as a rule felt that the care provided was more in line with the expectations they had of healthcare services, as they perceived fewer differences between their own explanatory model of illness and that of their Turkish physicians. That made them feel taken seriously, recognised as patients and treated as fully fledged individuals (Q11).

Q11: I notice when I'm in Turkey that they listen to me…. Okay, these are special clinics you visit, I understand that as well, but there they not only think about their own specialisation, but also outside of that…. It felt more like they took me seriously than they do here.

[I-03, 1rst generation migrant, paid work, no difficulty with Dutch language]

The respondents' narratives typically reflected a better match between their own explanatory models of illness and those of the Turkish healthcare system (Q12). Direct access to swift specialist diagnostics was seen as a way of shortcutting the stepwise diagnostics of Dutch GPs, in accordance with their felt urgency of risk reduction (Q13).

Q12: My [Dutch] GP, I told him something was wrong but I didn't know what. I told him it can't be good for your heart to beat at such a speed. He told me that's what it's made for. No, a heart is made to beat at a certain rhythm and to accelerate when you exercise or get frightened. It was made to function like that. I asked him what will happen if my heart goes on beating this fast? Yeah, then maybe you'll live 50 years instead of 80. You can't give a patient an answer like that. To me that was the limit. So I took a plane to Istanbul and went to a cardiologist because I thought something was wrong with my heart. The doctor thought I'd already had some bloodwork done, but he didn't ask what the results were. I told him they couldn't find anything, but that I have a high heart rhythm and I'm tired. He did all the diagnostics a cardiologist would do: an ultrasound, an exercise stress test, a bicycle test, a lab workup. In the ultrasound

they looked at the blood supply in my heart, looked for hardening of the arteries. Everything he told me put me at ease. He told me he could find nothing. He looked at my lab workup – he did a full workup – and he told me the problem was clear. He said they probably didn't test that. I had a vitamin D deficiency.

[I-03, 1rst generation migrant, paid work, no difficulty with Dutch language]

Q13: In the Netherlands they suspected my husband's pancreas, he was admitted to hospital, he had lost a lot of weight and they'd given him an appointment for a month later. He went to Turkey, got his results, went back and gave them the report. The answer he got was that they [the Dutch hospital] were also planning to look at that. But it takes a whole month, maybe it could have metastasised by then?

[I-09, 2nd generation migrant, unemployed, no difficulty with Dutch language]

Almost all respondents reported that more effective curative treatments were provided in Turkey. They implied that they perceived provision of strong medication or an operation to be the only two decisive curative options for health improvement, thereby shortcutting the stepwise treatment approach of the Dutch healthcare system (Q14).

Q14: In Turkey they tell me I need an operation for my most recent symptoms. Even if they tell me no in the Netherlands, I'll go to Turkey for the necessary [operation]. Nothing more I can do.

[I-02, 1.5 generation migrant, unemployed, difficulty with the Dutch language]

All but one respondent thus saw treatment in Turkey as a valid option. This was limited perhaps by a lack of sufficient holidays and/or lack of a local social network for aftercare, but definitely not by financial restraints, even if specific services were not reimbursable (Q15).

Q15: This year I was admitted to hospital [in Turkey] and they [Dutch insurance company] told me they might not pay for it. That doesn't interest me. Just let me know what's wrong and what we can do about it. I want to be free of the pain…. An MRI is just 400 lira, €50. I won't die from that expense, but getting an MRI in the Netherlands? … I also bring back a lot of medicines from Turkey nowadays.

[I-03, 1rst generation migrant, paid work, no difficulty with Dutch language]

### Healthcare utilisation on return to the Netherlands

On their return to the Netherlands, almost all respondents went back to their Dutch GP to present Turkish test results, diagnoses and/or suggested treatments in order to discuss subsequent steps in the treatment process. This was not always without difficulty (Q16).

Q16: I had great difficulty convincing my GP that this was it. Couldn't be possible! I showed him the blood levels. I don't understand them, but according to my GP my vitamin B was above the Dutch recommended level. But the professor in Turkey had told me … people from Turkey should have above 100…. I told my GP, listen, this and that is the case. She wouldn't believe it. But all right, it didn't matter, with great difficulty they gave me those 3 injections, once a week, and thank God, the symptoms were gone.

[I-05, 2nd generation migrant, paid work, no difficulty with Dutch language]

In case the consultation endpoint provided in Turkey was accepted by the Dutch healthcare provider, that justified the respondent's considerations for using Turkish health services and helped to compensate for perceived shortcomings in the Dutch system. At the same time, in case Turkish test results were not accepted, similar feelings of justification were expressed, as the fact that something had actually been found was perceived by respondents as comforting (Q17).

Q17: It's not nice to not be heard, that's what I notice in the Netherlands, be it a GP or a medical specialist. They're too simplistic about your issues, they send you away too easily and there's no alternative but to think up another solution…. Last time, my reassurance in Turkey was only because they do extensive testing. And even if it eventually turns out to be something small, they can explain my health issue with it. And if you treat that, it's over after that. That's my experience, that's why I prefer using healthcare in my country of origin. Nowadays I just buy a ticket without hesitating. My second opinion is now in Turkey.

[I-03, 1rst generation migrant, paid work, no difficulty with Dutch language]

Generally, respondents additionally argued that using healthcare in Turkey would be unnecessary and preventable if their GPs would just attend correctly to their health symptoms (Q18). Nonetheless, almost all respondents said they also appreciated the opportunity to discuss the test results and treatment options provided in Turkey on their return in the Netherlands. As this was seen as a specific quality of Dutch GPs—being a case manager for their patients' health and illnesses—the Turkish respondents continued to use primary as well as specialist healthcare in the Netherlands (Q19).

Q18: I had to go 3000 kilometres to learn that I have a vitamin D deficiency. No answer. 'We didn't think of that.' I told them my body was giving a signal. They told me it was all in my mind. I got vitamin D drops. I noticed my heart rhythm was decreasing. So that was the cause of my problem. You're inclined to start searching elsewhere. In the past 10 to 15 years I've been doing that more and more.

[I-03, 1rst generation migrant, paid work, no difficulty with Dutch language]

Q19: Also about my legs and sleeplessness. It had taken me years, and in Turkey the whole diagnosis took 2 hours and they told me I have restless legs. True, the whole night. He told me to take these pills, and thank God I still have those pills in case I need them. I went to the GP in the Netherlands and it all started up again from the beginning. Eventually she accepted it, but it had some side-effects. So again I'm happy with my GP, I can get along well with her, and we're now trying the third medication because of the side-effects…. Now when I get side-effects, we try something else. That's also positive…

[I-05, 2nd generation migrant, paid work, no difficulty with Dutch language]

However, sometimes when the Turkish and Dutch systems provided conflicting diagnoses or treatment options, respondents were left in despair, unsure which version to believe. These few respondents nevertheless continued to see transnational healthcare use as one option to find solutions to their health problems (Q20).

Q20: If everything is fine, why do I have problems? I'm still suffering…. I'm at the point where I am losing myself. They couldn't find a solution to my headaches. For years, I've been going to doctors here as well as in Turkey. If need be I'll just have to go in Turkey as well.

[I-02, 1.5 generation migrant, not employed, difficulty with Dutch language]

## DISCUSSION
### Summary of the findings
In the stories of Dutch residents with a Turkish ethnic background, this study observed how: (1) cross-border healthcare use was encouraged by cultural mismatches between expected and provided services and by differing explanatory models of illness upheld by patients and Dutch providers; (2) both transnationalism in patients and entitlements to insurance reimbursement facilitated the use of Turkish health services to bypass perceived barriers in the Dutch system; (3) cultural mismatches were reinforced during GP consultations after the patients' return to the Netherlands, thereby inducing further service use abroad.

### Interpretation
Our findings indicate that the regular transnational healthcare use of Dutch residents with a Turkish ethnic background[15] may be the result of mismatches between the patients' and physicians' explanatory models of illness in relation to cultural differences in both healthcare systems and national cultures. First, the feeling of our respondents that their illness sensations were not easily acknowledged as symptoms reflects that the interpretation of such sensations may heavily depend on culture.[22] Our findings and previous studies suggest that mismatches in explanatory models may result from a perceived cultural distance to the Dutch healthcare system[16] rather than from a perceived cultural distance to the Dutch society.[30] Second, such mismatches made some of our respondents feel a second-class citizen. In this respect, another Dutch qualitative study found that Dutch patients from Turkish origin regularly reported that they were being treated indifferently, inattentively or discriminatorily by physicians in primary care, while native Dutch patients did not mention such experiences.[31]

Despite the mismatches in explanatory models of illness, most of the respondents in our study continued to see their GP as their case manager and a reliable source in seeking care. This means that the GP in the Netherlands, and perhaps also in other counties with a similar gatekeeper system, could play key role in reducing transnational healthcare use. Although cultural gaps may be difficult to bridge,[26] cross-border healthcare utilisation also rests on individual beliefs, motivation and decisions.[20 21] Therefore, one approach could be to strengthen patient centred primary care,[32] by providing culturally sensitive and competent care during the consultation, as to achieve mutual understanding between the patient and the GP.[22 33] First, this includes being aware of cultural differences, such as in uncertainty avoidance,[28] and the related need to find pathophysiological causes of sensations and symptoms. Second, it requires cross-cultural communication,[33] such as discussing mismatches and negotiating the need for transnational healthcare use,[22] including possible iatrogenic outcomes and timely seeking professional help if these might occur.[5]

However, providing culturally sensitive care alone may not be sufficient to change the transnational healthcare use of Dutch residents of Turkish origin. Our results confirm that the insurers' reimbursement of healthcare and/or the low cost of care in the country of origin may play an equally important part in transnational healthcare uptake.[1 34] These instruments made opting for health services in Turkey feel natural to our respondents. For Dutch insurance companies, one option to limit the higher than average healthcare use could be contracting only a selection of care providers for the delivery of care to people with Dutch insurance.[13] Currently, insurance companies reimburse services from all possible providers abroad, which can be justified by comparable scores on quality outcome indicators,[25] while in the Netherlands they only reimburse care from 'preferred providers' meeting additional quality process standards.[26] Introducing similar quality procedures for the delivery of healthcare in countries abroad could also contribute to a further harmonisation between services in different countries as well as to cost-effective care, while the patient's option to obtain care abroad is being maintained. To additionally minimise the risks of medicalisation and adverse iatrogenic effects, and to coordinate transnational diagnostics and treatment, an online portal

Şekercan A, et al. BMJ Open 2021;11:e051903. doi:10.1136/bmjopen-2021-051903

for international medical information transfer could be considered, although achieving personal data safety in conformity with privacy legislations may be challenging.

## Strength and limitations

Our main strength is the use of a cultural systems perspective.[22 35] This helped us to see how transnational healthcare use may be encouraged by an interaction of factors in both clinical practice, healthcare system and national culture. Another strength lies in the ethnic-concordant interviewer (AŞ), who was familiar with respondents' language and cultural expressions. As the biographic-narrative interpretive method[29 36 37] led to in-depth and meaningful information, we feel confident to have successfully addressed the potential downsides of such an interviewer, by both reassuring anonymity and letting respondents have ownership of the interviews.

An important limitation is the rather small number of participants in the study, sampled from two primary care practices, in one transnational healthcare context. This limits the possibility to generalise our findings to other Dutch primary care patients of Turkish background as well as to other patient populations, both within and beyond our country. Although the pattern in transnational healthcare use we found was pronounced, additional research in other populations and transnational healthcare contexts seems warranted.

A second limitation is that we only included patients with a Turkish background and no patients of Dutch ethnic origin. Therefore, we cannot say that it is especially Dutch patients of Turkish origin that bypass the GP to directly access specialist care. Similarly, we cannot tell whether the Dutch patients of Turkish origin, while visiting Turkish healthcare services, were treated differently from the Turkish residents without a migration history. As patient satisfaction can increase when such a bypass option is available,[38] it would be interesting to compare our findings with data from patients of Dutch ethnic origin.

## CONCLUSIONS

Our study uncovered how transnational healthcare use of Dutch patients with a Turkish background was encouraged by mismatches with their Dutch GP. Their dissimilar explanatory models of illness reflected differences in national healthcare systems and national cultures. In order to reduce the unwelcome consequences of transnational healthcare use, measures may include strengthening the provision of culturally sensitive care in the country of residence and restricting the reimbursement of care in the country origin while maintaining the option to obtain care abroad. Future research could include other migrant populations, qualitatively compare the healthcare experiences of migrant and native communities, and quantitatively assess the importance of the patterns we identified.

**Acknowledgements** We are most grateful for the investments of time and energy made by the respondents in sharing their stories and welcoming the first author into their homes. Without them, this study would not have been possible.

**Contributors** AŞ collected the data, analysed the interviews and drafted the manuscript. JH read the coded interviews and reflected on the analysis and interpretation of the data. KS and RJGP contributed to the conception and design of the study. KS also provided the daily supervision of the fieldwork. All authors were involved in revising initial drafts of the manuscript, and read and approved the final manuscript.

**Funding** The first author was a PhD candidate at the Amsterdam UMC, University of Amsterdam, with research funded by a MD-PhD fellowship from the same institution. Grant number: N/A.

**Competing interests** None declared.

**Patient consent for publication** Not applicable.

**Ethics approval** According to the Dutch Medical Research Involving Human Subjects Act, this interview study did not require approval by a medical research ethics committee. Data and analysis logs were stored in a protected digital environment, in accordance with Amsterdam University Medical Centers guidelines and Dutch and EU privacy legislation.

**Provenance and peer review** Not commissioned; externally peer reviewed.

**Data availability statement** No data are available. The interview transcripts and the data analysis in the MaxQDA project file are not publicly available. In order to build the required trust, the respondents' informed consent remained restricted to the use of the interview data for the present study only.

**ORCID iD**
Janneke Harting http://orcid.org/0000-0002-2811-9551

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
