## [Reviewer comments · BMJ Open]

ARTICLE DETAILS

TITLE (PROVISIONAL)	Understanding transnational healthcare use in immigrant communities from a cultural systems perspective: A qualitative study of Dutch residents with a Turkish background
AUTHORS	Şekercan, Aydin; Harting, Janneke; Peters, Ron; Stronks, Karien

VERSION 1 – REVIEW

REVIEWER	Harris, Mark University of New South Wales, School of Public Health and Community Medicine
REVIEW RETURNED	09-May-2021

GENERAL COMMENTS	This is an interesting small qualitative study of Dutch patients of Turkish origin. The focus on transnational health care and the reasons behind this. There are a number of issues that warrant attention, however. 1. The authors use the term “sensations of illness”. This needs to be defined as it is not in usual use. Although symptoms are technically “a physical or mental feature which is regarded as indicating a condition of disease” in common use they do not necessarily need to be linked to a diagnosed condition.2. The methods section includes paragraphs comparing Turkish and Dutch healthcare systems and cultures (Line 15-57 page 7 and lines 1-18 page 8). These would more properly belong in the introduction.3. The sample of 13 participants is very heterogeneous especially with regards migration generation and Netherlands born. The analysis implies that responses were homogenous. Were there differences in the perceived drivers within the sample? (For example was there a difference between migration generation groups?)4. The presentation of quotes is a little difficult to read. Have the authors considered including them in the text?5. In the discussion the authors state that “cross-border healthcare use was fostered by cultural mismatches..” . Given the nature of the evidence “was fostered” should be replaced with “was associated with”, as causation was not evaluated. This is repeated in the conclusion.6. In discussing cultural differences, it would be useful to discuss differences in the cultural meaning of “sensations” or symptoms especially related to mental health and the impact of acculturation on different generations (cf Fassaert T et al. Acculturation and psychological distress among non-Western Muslim migrants—A population-based survey. Int. J. Soc. Psychiatr. 2011;57:132–143).7. It would also be useful to discuss the possibility that doctors may be treating these patients differently. In the Dutch context the possibility of prejudice clinicians should be discussed. Similarly in
--

	the Turkish context it would be useful to discuss the possibility that doctors are treating expatriates differently from Turkish residents because they were in country for only a brief period of time and they felt pressured to assess and treat quickly. 8. The paragraph on system changes needed (lines 13-47 page 17) makes a number of assumptions. The most important is that Turkish doctors do not meet the quality standards of Dutch doctors. This may be true especially with respect to continuity of care in family practice (Lionis C. Primary care in Greece. 2011. http://www.euprimarycare.org/column/primary-care-greece). However it needs to be discussed carefully with reference to evidence. 9. The authors do note the lack of comparison with Dutch patients as a limitation (there is also a grammatical effort in the second sentence Line 35 page 18). This is an important limitation and further research making comparison should be recommended in the conclusion.
--	--

REVIEWER	Durham, Jo Queensland University of Technology, School of Public Health & Social Work
REVIEW RETURNED	10-May-2021

GENERAL COMMENTS	Thank you for the opportunity to review this paper, it is generally well-written and easy to read but I feel it lacks depth of analysis and interpretation There are also some minor errors and the manuscript should be edited Introduction In the introduction a brief sentence/clause explaining what is meant by medicalization would be helpful especially as this is one of the main reasons the authors believe transnational healthcare use might be harmful. To better justify the why we need to be concerned about medicalization adverse consequences as well as iatrogenic diseases can you add other potential negative outcomes for the health system and more clearly make the link between travelling overseas for health care and medicalization (and evidence to support your claim)? There is a rich literature contemporary transnational healthcare (sometimes called 'medical tourism', 'medical migration', 'international healthcare consumption'), where patients cross border for care because people feel their country of residence does not provide the care they need it they which could be briefly discussed A bit more detail about the Dutch healthcare system and the Turkish one would be useful especially related to financing, I'm not clear if people using the Dutch and Turkish healthcare services are using public or private services in each location. If I understand right patients are insured by private Dutch firms which routinely also cover costs of treatment overseas (i.e., not travel insurance)? The discussion on page 7 of the cultural differences is useful although I think in considering national cultures its important not to stereotype and to also recognise that culture is dynamic Is there any data on what people travel to Turkey for regarding healthcare? Can you add a brief portrait of the demographics of
--

	Turkish background in the Netherlands and why they may use health services more (especially compared to co-ethnics)? Are they residents or citizens? Are they dual citizens? Methods The methods are generally clear What made you decide to introduce the diagram of a human life cycle? How did it change the responses? Using the human life cycle how did you also step participants through the different phases of the consult? What language were the interviews conducted in? Were the interviews transcribed? By whom if so? Was any translation required from one language to another? How was this conducted if so? The sample size is a limitation Findings I feel more depth and interpretation can be added in each of the themes. I also found referring to table for the quotes a bit cumbersome. Pg. 13 – “. They felt that both basic physical examination and lab testing were definitely needed to find out the cause” – do you mean everyone? Pg 14 - where people believed unnecessary harm or death could have been prevented, . . – how many people believed hard could have been prevented? More nuance is needed to some of the claims /quotes – for example when you only give one point to illustrate a point is the quotes reflective of more than the one person who provided the quote It would also be helpful to know who made the quote – e.g an older male respondent who frequently travelled to Turkey for healthcare noted Where did participants go in Turkey? To major cities /urban/rural areas? Where thy had family? Many is very vague and as you have small sample could say 3 of the 12 , . . . Q12 shows how taking someone seriously and doing all the necessary tests was needed (esp given Vit D deficiency is not uncommon) – is Q18 from the same person or did more than one person have vit d deficiency? Q15 shows the patient also travels because he/she values their health – e.g. they said they would go even if not reimbursed – this reinforces potentially the common-sense decision to go to Turkey to get a proper diagnosis and reduce uncertainty and fear Why did the person in Q15 bring back lots of medicines from turkey – are they cheaper? Considered more effective? Did participants use the public or private system I found “It not nice to be heard” particularly powerful and this while not stated seems to come across ion several of the quotes and suggests a feeling of being disenfranchised More detail will help to understand and contextualise the quotes and interpretation
--	---

	Presenting the quotes by a number in chronological order also makes it hard to know how many participants are you quoting? It could be one! Can you use the code you gave each participant you quote and provide some demographic information? Discussion I'm not sure you demonstrated reimbursement and cheaper cost of care are drivers of using HC in Turkey Line 57 p. 16 "cultural differences" – should this be "differences in explanatory models of diseases and expectations" – or something similar, "cultural differences" can perpetuate stereotypes, and ignore the dynamic nature of culture. Isn't it more important that GPs take a client centred approach and listen to and respecting the needs and preferences of their clients, expressing empathy, work with uncertainty etc? Consider for the example the participants who said he/she felt "no heard" - there may however be structural constraints to providing such patient centred care (e.g standard consultation time etc. Nevertheless, approaching consultations as social encounters, not just medical ones, would give GPs a better understanding of how users experienced primary healthcare Any potential ethical issues or other challenges with an online portal/international medical information transfer? Is there evidence from your findings transnational healthcare is harmful?
--	---

VERSION 1 – AUTHOR RESPONSE

Reviewer: 1

Prof. Mark Harris, University of New South Wales Comments to the Author:

This is an interesting small qualitative study of Dutch patients of Turkish origin. The focus on transnational health care and the reasons behind this. There are a number of issues that warrant attention, however.

1. The authors use the term "sensations of illness". This needs to be defined as it is not in usual use. Although symptoms are technically "a physical or mental feature which is regarded as indicating a condition of disease" in common use they do not necessarily need to be linked to a diagnosed condition.

We agree that readers may not be familiar with the term "sensation". Hay (2008) argues that the transformation of a "sensation", as the basis of our recognition that something is wrong, into a "symptom", as a recognized objective clinical reality, is an essential element of the first phase of a medical consultation. In the methods section, we've now added Hay's definitions of the respective terms, together with the reference on which they are based (Analytical framework).

In the first phase, a patient presents his or her sensations, as the basis of the recognition that something is wrong, and tries to persuade the healthcare provider to transform these sensations into symptoms, as a recognized objective clinical reality (24).

2. The methods section includes paragraphs comparing Turkish and Dutch healthcare systems and cultures (Line 15-57 page 7 and lines 1-18 page 8). These would more properly belong in the introduction.

We thank the reviewer for this suggestion, and we've reconsidered the position of this paragraph. However, we preferred to keep it in the methods section, as this comparison was part of our framework. To make this more explicit, in the method section we changed the subheading into 'analytical framework', and added a short introduction to this framework.

In the case of transnational healthcare use, cultural mismatches between explanatory models are more likely to occur, as a medical consultation is influenced by the rules and values of two national healthcare systems and two national cultures. Therefore, the analytical framework comprised the concept of cultural mismatches, the differences between national health care systems, and a comparison of national cultures.

3. The sample of 13 participants is very heterogeneous especially with regards migration generation and Netherlands born. The analysis implies that responses were homogenous. Were there differences in the perceived drivers within the sample? (For example was there a difference between migration generation groups?)

The sample was indeed very heterogeneous, while the patterns we found in their stories were not. We thank the reviewer for making clear that we not explicitly mentioned the homogeneity of the pattern of drivers we found. Although respondents sometimes differently interpreted the mismatches they experienced (e.g., as the physician lacking knowledge or as the physician treating migrant patients differently), such differences were not related to characteristics of the respondents, such as migration generation. In response to this comment, in the data analysis section we now refer to the homogeneity of the drivers behind transnational healthcare that were reported.

Emerging themes and patterns were visualised and discussed together. The pattern of drivers behind transnational health care use was almost homogeneous across the sample. Both authors deemed data saturation to have been reached after analysing nine interviews.

4. The presentation of quotes is a little difficult to read. Have the authors considered including them in the text?

We opted for a presentation of quotes in tables, as to meet the requirement of the journal in terms of maximum word count, but we agree with the reviewer that presenting quotes close by the text they are meant to illustrate will improve the readability of the manuscript. Therefore, in the revised version we included the quotes in the results section. As this did not change the content of our work, we did not mark them in the revised manuscript.

5. In the discussion the authors state that "cross-border healthcare use was fostered by cultural mismatches..". Given the nature of the evidence "was fostered" should be replaced with "was associated with", as causation was not evaluated. This is repeated in the conclusion.

We agree that causal relations cannot be drawn from our qualitative study. In the revised manuscript (abstract, summary of the findings, conclusions) we make clear that this – and some other – "causalities" reflect the experience or explanations of the respondents rather than our inferences. Here, we present one example of such a change.

In the respondents' stories, we observed how: (A) cross-border healthcare use was encouraged by cultural mismatches between expected and provided services and by differing explanatory models of illness upheld by patients and Dutch providers;

6. In discussing cultural differences, it would be useful to discuss differences in the cultural meaning of "sensations" or symptoms especially related to mental health and the impact of acculturation on different generations (cf Fassaert T et al. Acculturation and psychological distress among non-Western Muslim migrants—A population-based survey. *Int. J. Soc. Psychiatr.* 2011;57:132–143).

We thank the reviewer for this suggestion. In our first interpretation paragraph, we now acknowledge that the interpretation of sensations might heavily depend on culture, we refer to the work of Kleinman in this respect. The work of Fasseart provide some further insight in the source of the culture-driven mismatches in explanatory models of disease.

First, the feeling of our respondents that their illness sensations were not easily acknowledged as symptoms reflects that the interpretation of such sensations may heavily depend on culture (22). Our findings and previous studies suggest that mismatches in explanatory models may result from a perceived cultural distance to the Dutch healthcare system (16) rather than from a perceived cultural distance to the Dutch society (30).

7. It would also be useful to discuss the possibility that doctors may be treating these patients differently. In the Dutch context the possibility of prejudice clinicians should be discussed. Similarly in the Turkish context it would be useful to discuss the possibility that doctors are treating expatriates differently from Turkish residents because they were in country for only a brief period of time and they felt pressured to assess and treat quickly.

Again a very useful suggestion, which we adopted by connecting our findings to those of another Dutch qualitative study in this respect.

Second, such mismatches made some of our respondents feel a second-class citizen. In this respect, another Dutch qualitative study found that Dutch patients from Turkish origin regularly reported that they were being treated indifferently, inattentively or discriminatorily by physicians in primary care, while native Dutch patients did not mention such experiences (31).

We agree with the reviewer's suggestion that our respondents, while making use of healthcare in Turkey, could have been treated differently from Turkish residents. We have included in our limitations that we do not know whether this was indeed the case.

Similarly, we cannot tell whether the Dutch patients of Turkish origin, while visiting Turkish health care services, were treated differently from the Turkish residents without a migration history.

8. The paragraph on system changes needed (lines 13-47 page 17) makes a number of assumptions. The most important is that Turkish doctors do not meet the quality standards of Dutch doctors. This may be true especially with respect to continuity of care in family practice (Lionis C. Primary care in Greece. 2011. <https://eur04.safelinks.protection.outlook.com/?url=http%3A%2F%2Fwww.euprimarycare.org%2Fcolumn%2Fprimary-care-greece&data=04%7C01%7Cj.harting%40amsterdamumc.nl%7C3358bf4a17c7414e894308d921f2f270%7C68dfab1a11bb4cc6beb528d756984fb6%7C0%7C0%7C637578149598638002%7CUnkno>

wn%7CTWFpbGZsb3d8eyJWIjoiMC4wLjAwMDAiLCJQIjoiV2luMzliLCJBTiI6Ikk1haWwiLCJXVCI6Mn0%3D%7C1000&sdata=41I8sWLk3g2urIkLh37JWO0Ex3Pf6HtVHSb3C0gArnU%3D&reserve d=0).

However it needs to be discussed carefully with reference to evidence.

We were a little surprised by the reviewer's observation, but after re-reading this paragraph, we have to admit that it could be understood as an assumption that Turkish doctors would not meet the quality standards of Dutch doctors. However, this is not the message that we intended to send, as in our method section we presented the Turkish and Dutch healthcare systems as providing "similar quality of care in objective terms, but they differ in the organisation and delivery of services". In the revised discussion, we have made a clear distinction between quality of care in terms of outcome indicators (which is comparable for both countries) and quality process indicators, that additionally could be used as an instrument to regulate transnational healthcare use.

Currently, insurance companies reimburse services from all possible providers abroad, which can be justified by comparable scores on quality outcome indicators (25), while in the Netherlands they only reimburse care from "preferred providers" meeting additional quality process standards (26). Introducing similar quality procedures for the delivery of healthcare in countries abroad could also contribute to a further harmonisation between services in different countries as well as to cost-effective care, while the patient's option to obtain care abroad is being maintained.

9. The authors do note the lack of comparison with Dutch patients as a limitation (there is also a grammatical effort in the second sentence Line 35 page 18). This is an important limitation and further research making comparison should be recommended in the conclusion.

We thank the reviewer for this comment. We corrected the grammatical error, and included the suggested recommendation in the concluding section of the revised manuscript.

Future research could include other migrant populations, qualitatively compare the healthcare experiences of migrant and native communities, and quantitatively assess the importance of the patterns we identified.

We also corrected the grammatical errors the reviewer observed. This paragraph now reads as follows.

Therefore, we cannot say that it is especially Dutch patients of Turkish origin that bypass the general practitioner to directly access specialist care.

Reviewer: 2

Dr. Jo Durham, Queensland University of Technology Comments to the Author:

Thank you for the opportunity to review this paper, it is generally well-written and easy to read but I feel it lacks depth of analysis and interpretation There are also some minor errors and the manuscript should be edited

We thank the reviewer for challenging us to add further depth to the analysis and interpretation. In response to the reviewer's comments below, we've incorporated more precision in our results section and additional points in our discussion section.

Introduction

In the introduction a brief sentence/clause explaining what is meant by medicalization would be helpful especially as this is one of the main reasons the authors believe transnational healthcare use might be harmful. To better justify the why we need to be concerned about medicalization adverse consequences as well as iatrogenic diseases can you add other potential negative outcomes for the health system and more clearly make the link between travelling overseas for health care and medicalization (and evidence to support your claim)?

We agree that we could further substantiate the background of the study. In the revised manuscript, we have included a brief definition of medicalisation, and added various potential negative outcomes, including some at the level of the health system, in order to strengthen the link between medicalisation and transnational healthcare use.

High transnational healthcare use may increase the risk of medicalisation, defined as expansion of the medical domain (4).

In addition, transnational healthcare use is likely to foster antimicrobial drug resistance (6). At the system level, it may increase healthcare spending in the country of residence, pull resources from the public sector in the country of origin, and thus widen health disparities (6). Motivators for treatment abroad generally include low costs, short waiting lists, quality of care and the available medical procedures (8, 9). Yet, the drivers that underlie frequent transnational healthcare use are complex and deserve further investigation(9), as to curtail its adverse effects.

There is a rich literature contemporary transnational healthcare (sometimes called 'medical tourism', 'medical migration', 'international healthcare consumption'), where patients cross border for care because people feel their country of residence does not provide the care they need it they which could be briefly discussed

We thank the reviewer for this suggestion. We've included a brief summary of what is known about the topic of transnational healthcare use in the first paragraph of the introduction section. This summary was already presented in response to the previous comment.

A bit more detail about the Dutch healthcare system and the Turkish one would be useful especially related to financing, I'm not clear if people using the Dutch and Turkish healthcare services are using public or private services in each location. If I understand right patients are insured by private Dutch firms which routinely also cover costs of treatment overseas (i.e., not travel insurance)?

We understand the need for some more information. In our background, we have now included a brief explanation about relevant EU regulations and coverage of healthcare by Dutch insurance companies.

Being residents of the European Union, they have access to necessary health care in all countries in Europe (12) and Dutch insurance companies reimburse public as well as private healthcare in both the Netherlands and Turkey (13).

The discussion on page 7 of the cultural differences is useful although I think in considering national cultures its important not to stereotype and to also recognise that culture is dynamic

In our description of cultural differences, we did not meant to stereotype. The distinctions we mention are based on empirical research. In the revised manuscript, we have now included the reference

[Hofstede, 2001] for every difference we present. Still, we decided it would have added value to comment on a cultures dynamic in the text too.

In terms of national cultures, Turkey has a more hierarchical society than the Netherlands (28). In such a society, power, authority and control are often centralised (physicians make the decisions, for example). In addition, collectivity is more important in the Turkish than in the Dutch culture(28). That may imply, for instance, that the family is more involved in the healthcare utilisation trajectory and that the provision of care is more family-based. A final difference in national cultures is the higher level of uncertainty avoidance in Turkey (28). This means that members of the society feel a greater degree of discomfort with uncertainty and ambiguity. This may translate into a stronger desire to rule out health risks and potential diseases and to rule in some pathophysiological cause to explain sensations or symptoms. Despite these differences between the two nations, it should be noted that culture is a dynamic concept, that may express itself in different ways.

Is there any data on what people travel to Turkey for regarding healthcare? Can you add a brief portrait of the demographics of Turkish background in the Netherlands and why they may use health services more (especially compared to co-ethnics)? Are they residents or citizens? Are they dual citizens?

We understand that the reviewer prefers some additional information about the Dutch residents with Turkish background who frequently make use of healthcare in the country of origin. In the background, we included a brief portrait of the migrant from Turkey. This also explains that they are residents of whom most identify as both Turkish and Dutch. Regarding their healthcare use, we have also added the few relevant details we have the disposal of.

With about 425.000 people, they form the largest ethnic minority group in the Netherlands (10). Migration from Turkey was encouraged in the 1960s and early 1970s to fill labour shortages in unskilled occupation (11). In the years that followed (1970-80), many “guest workers” brought their spouses and children to the Netherlands (11). As 80% of the Dutch residents of Turkish background identify as being both Dutch and Turkish and maintain ties with the Turkish culture, they can readily compare, utilise and evaluate various features of health services in both countries (2, 7).

Their overall rates of utilisation are also high in comparison with their co-ethnics who do not obtain healthcare in Turkey (Dutch primary care use: 40.4% versus 22.5%; specialist medical care use: 57.8% versus 33.9%) (15, 17). The few factors identified to be associated with healthcare use in the country of origin are poorer self-reported health and a wider perceived cultural distance to the healthcare system in the country of residence (16).

Personal motives underlying the use of the health services in Turkey were already explained in the third paragraph of the introduction section. Finding further explanations for this use was the aim of the present study.

Methods

The methods are generally clear

We thank the reviewer for this positive evaluation.

What made you decide to introduce the diagram of a human life cycle? How did it change the responses?

The introduction of the diagram of the human life cycle was triggered by the interviewer's experience that respondents tended to focus on one episode of healthcare use or to discuss different episode while mixing these up. We have now added this reason for using the life cycle to the interview approach in de method section.

In the first three interviews the respondents tended to focus on one episode of healthcare use or to discuss different episodes while mixing these up. Therefore, in the following interviews the interviewer began showing respondents a diagram of a human life cycle, so as to visually guide them in retrieving their stories in a chronological order without interfering in their thought-forming processes.

Using the human life cycle how did you also step participants through the different phases of the consult?

The different phases of the consult were subject of the second part of the interview, in which the interviewer asked the respondents to elaborate on the topics they had raised in the first part of the interview. We have now also included this in the interview approach, with some further explanation in Supplementary Material I. Interview guide.

In the second part of the interview, while sticking to the respondent's order of topics raised and their choice of words used, the interviewer asked the respondent to elaborate on or clarify certain narratives, specifically in relation to the different phases of the consultation. For the interview guide, please see Supplementary Material I.

What language were the interviews conducted in?

The interviews were primarily held in Turkish, although some eventually became mixtures of Turkish and Dutch. This information can be found in the data collection paragraph.

Were the interviews transcribed? By whom if so? Was any translation required from one language to another? How was this conducted if so?

The interviews were transcribed and directly translated – if necessary – from Turkish in English by the first author. This information can be found in the data collection paragraph.

The sample size is a limitation

We certainly agree with the reviewer that the sample size is a limitation. Therefore, we addressed this issue in the strength and limitations section of the discussion, as well as in the "article summary" preceding the main text of the manuscript.

Findings

I feel more depth and interpretation can be added in each of the themes.

We agree that the findings could benefit from more depth and interpretation, and have taken this comment to heart. Please see our response to the detailed comments below.

I also found referring to table for the quotes a bit cumbersome.

We opted for a presentation of quotes in tables, as to meet the requirement of the journal in terms of maximum word count, but we agree with the reviewer that presenting quotes close by the text they are meant to illustrate will improve the readability of the manuscript. Therefore, in the revised version

we included the quotes in the results section. As this did not change the content, we've not marked these changes in the revised manuscript.

Pg. 13 – “. They felt that both basic physical examination and lab testing were definitely needed to find out the cause” – do you mean everyone?

Pg 14 - where people believed unnecessary harm or death could have been prevented, . . – how many people believed hard could have been prevented?

We've now included a clearer indication of the proportion of respondents supporting the various patterns in the results section of the manuscript. We think this further illustrates the homogeneity of the pattern in drivers for transnational healthcare use we observed in the stories of our respondents. Still, as this is quite uncommon in qualitative research, we do not report the precise number of respondents. Here we present one example of the clearer indications we included throughout the results section.

In the first phase of consultation, respondents typically felt that their general practitioner chose a wait-and-see approach, and they often did not understand why. They felt that their sensations were either not acknowledged as symptoms, or were interpreted as symptoms giving insufficient reason for further diagnostic investigation to rule out underlying causes. This approach was out of line with the perceived urgency of the respondents' embodied sensations and their suspicions that something was wrong. All but one of the respondents explained that both basic physical examination and lab testing were definitely needed to find out the cause (Q01). The vast majority said they tried numerous times to convince their GPs to deviate from the wait-and-see approach, often without success (Q02).

More nuance is needed to some of the claims /quotes – for example when you only give one point to illustrate a point is the quotes reflective of more than the one person who provided the quote It would also be helpful to know who made the quote – e.g an older male respondent who frequently travelled to Turkey for healthcare noted

We appreciate the suggestion of the reviewer. For each quote, next to the 'interview number', we included the respondents migration status (first, 1.5 or second), employment status (e.g., paid work or unemployed), and difficulty with the Dutch language (yes/no). We choose these characteristics given their potential to influence transnational healthcare use. We did not include more than three characteristics, as to protect the respondent from the risk of becoming recognized. This revision also makes clear that we quoted in total 7 out of the 13 respondents. Using one quote to illustrate a collective finding and selecting respondents that expressed themselves best, are not uncommon methods of working in qualitative research. Here we present one example of the changes we made.

[I-03, 1st generation migrant, paid work, no difficulty with Dutch language]

Where did participants go in Turkey? To major cities /urban/rural areas? Where they had family?

We did not make an inventory of the geographical location the respondents had went to in Turkey, but we have added some information on the primary reason for their travel, their decision to make use of healthcare services, and the type of services they used (first paragraph in the results, respondents' characteristics).

Our 13 participants were between 39 and 78 years of age; 5 were in paid employment, 6 were born in the Netherlands, and 5 reported difficulty with the Dutch language (Table 1). Their primary reason to go to Turkey was visiting relatives, while decisions about using healthcare were in general made after

arriving in Turkey. Respondents usually made use of private healthcare services, as these were regarded more accessible and providing quicker services than public services.

Many is very vague and as you have small sample could say 3 of the 12 , .. .

Please see our response above.

Q12 shows how taking someone seriously and doing all the necessary tests was needed (esp given Vit D deficiency is not uncommon) – is Q18 from the same person or did more than one person have vit d deficiency?

We agree that taking someone seriously is needed. We think this observation is sufficiently discussed in the revised discussion of our manuscript, which now additionally includes a brief discussion of treating patients from Turkish origin differently than patients from Dutch origin, in response to one of the other comments of reviewer 2 – please see below.

Q15 shows the patient also travels because he/she values their health – e.g. they said they would go even if not reimbursed – this reinforces potentially the common-sense decision to go to Turkey to get a proper diagnosis and reduce uncertainty and fear

We tend to agree with the reviewer that this quote supports potentially the common sense decision to go to Turkey. However, in our results section we prefer to refrain from evaluative statement like this one, and to present the findings from the perspective of the patients. Therefore, just before Q15, we more neutrally summarized this “common-sense decision” as “Most respondents thus saw treatment in Turkey as a valid option.”

Why did the person in Q15 bring back lots of medicines from turkey – are they cheaper? Considered more effective?

We do not exactly know the answer to this question, as the respondent did not explain the reasons for bringing back lots of medicines from Turkey. Although this might have to do with the lower prices and more varied availability (nowadays Dutch pharmacies have the obligation to deliver the cheapest products), we prefer not to include this interpretation in the results section. Also, we think this issue is too detailed for further discussion.

Did participants use the public or private system

We thank the reviewer for this question. In Turkey, the participants usually made use of the private healthcare services. We included this information in the respondents characteristics in the result section.

Their primary reason to go to Turkey was visiting relatives, while decisions about using healthcare were in general made after arriving in Turkey. Respondents usually made use of private healthcare services, as these were regarded more accessible and providing quicker services than public services.

I found “It not nice to be heard” particularly powerful and this while not stated seems to come across in several of the quotes and suggests a feeling of being disenfranchised

We thank the reviewer for this valuable observation. In response, we included a brief discussion with respect to this finding in the first paragraph of our interpretation section in the discussion.

Second, such mismatches made some of our respondents feel a second-class citizen. In this respect, another Dutch qualitative study found that Dutch patients from Turkish origin regularly reported that they were being treated indifferently, inattentively or discriminatorily by physicians in primary care, while native Dutch patients did not mention such experiences (31).

More detail will help to understand and contextualise the quotes and interpretation Presenting the quote3s by a number in chronological order also makes it hard to know how many participants are you quoting? It could be one! Can you use the code you gave each participant you quote and provide some demographic information?

Please see our responses above. We have included the quotes throughout in the result section and added three relevant respondent characteristics to each of the quotes.

Discussion

I'm not sure you demonstrated reimbursement and cheaper cost of care are drivers of using HC in Turkey.

We assume that the reviewer is pointing at our summary of the findings, the first paragraph in our discussion. We agree that we did not “demonstrate” this relationship, as we based this the perception the respondents shared with us in the interviews. Therefore, we change the first sentence of this paragraph accordingly.

In the stories of Dutch residents with a Turkish ethnic background, this study observed how: [...]

Similar qualifications with respect to our findings were included in the abstract and the conclusion of the revised manuscript.

Line 57 p. 16 “cultural differences” – should this be “differences in explanatory models of diseases and expectations” – or something similar, “cultural differences” can perpetuate stereotypes, and ignore the dynamic nature of culture.

We thank the reviewer for this comment. We agree that the differences in explanatory models of illness should be more central to our argument, but we prefer to keep the addition that such differences might be related to cultural differences in both healthcare systems and national cultures, as this is in conformity with the analysis of the interview. We rephrased the first sentence of the interpretation section as follows.

Our findings indicate that the regular transnational healthcare use of Dutch residents with a Turkish ethnic background (15) may be the result of mismatches between the patients' and physicians' explanatory models of illness in relation to cultural differences in both healthcare systems and national cultures.

Accordingly, we rephrased the first sentence of the concluding paragraph.

Our study uncovered how transnational healthcare use of Dutch patients with a Turkish background was encouraged by mismatches with their Dutch general practitioner. Their dissimilar explanatory models of illness reflected differences in national healthcare systems and national cultures.

Isn't it more important that GPs take a client centred approach and listen to and respecting the needs and preferences of their clients, expressing empathy, work with uncertainty etc? Consider for the example the participants who said he/she felt "no heard" - there may however be structural constraints to providing such patient centred care (e.g standard consultation time etc. Nevertheless, approaching consultations as social encounters, not just medical ones, would give GPs a better understanding of how users experienced primary healthcare

We agree first of all that respondents' expressions like "It's not nice to not be heard" deserve a more elaborate discussion. We have now included this in the first paragraph of the interpretation section.

Second, such mismatches made some of our respondents feel a second-class citizen. In this respect, another Dutch qualitative study found that Dutch patients from Turkish origin regularly reported that they were being treated indifferently, inattentively or discriminatorily by physicians in primary care, while native Dutch patients did not mention such experiences (31).

Second, we think the reviewer is right in stating that preventing mismatches in explanatory models of illness primarily requires a patient centred approach. However, in case of a culture-related basis for such mismatches, we argue that patient centred also means a culturally sensitive approach including culturally sensitive case. In what now had become the second paragraph of the interpretation section, we refer to a patient centred approach, while operationalizing this as a culturally sensitive one.

Therefore, one approach could be to strengthen patient centred primary care (32), by providing culturally sensitive and competent care during the consultation, as to achieve mutual understanding between the patient and the GP (22, 33).

Any potential ethical issues or other challenges with an online portal/international medical information transfer?

We agree with the reviewer that it might be challenging to sufficiently protect international medical information transfer as to warrant personal data safety in conformity with privacy legislation. We added this potential challenge to the final sentence of the interpretation section.

To additionally minimise the risks of medicalisation and adverse iatrogenic effects, and to coordinate transnational diagnostics and treatment, an online portal for international medical information transfer could be considered, although achieving personal data safety in conformity with privacy legislations may be challenging.

Is there evidence from your findings transnational healthcare is harmful?

Please see our previous response to the reviewer's request to elaborate on the phenomenon of transnational healthcare. We included the most prominent harms in the first paragraph of the introduction section.

In addition, transnational healthcare use is likely to foster antimicrobial drug resistance (6). At the system level, it may increase healthcare spending in the country of residence, pull resources from the public sector in the country of origin, and thus widen health disparities (6). Motivators for treatment abroad generally include low costs, short waiting lists, quality of care and the available medical procedures (8, 9). Yet, the drivers that underlie frequent transnational healthcare use are complex and deserve further investigation(9), as to curtail its adverse effects.

VERSION 2 – REVIEW

REVIEWER	Harris, Mark University of New South Wales, School of Public Health and Community Medicin
REVIEW RETURNED	29-Jun-2021
GENERAL COMMENTS	The authors have adequately addressed all the issues raised in my previous review.